# NDIM: Neuronal Diversity Inspired Model for Multisensory Emotion Recognition

## Abstract

Without cross-sensory interaction, a key aspect of multisensory emotion recognition, traditional deep learning methods exhibit inferior performance in this task. On the contrary, the human brain possesses an inherent and remarkable capacity for multisensory recognition. Its diverse neurons exhibit distinct responses to sensory inputs, thus facilitating cross-sensory interaction. Leveraging this superiority, we propose the Neuronal Diversity Inspired Model (NDIM), which incorporates both unisensory and multisensory neurons, aligning with the human brain. To mirror the diverse response characteristics exhibited by various neurons, we introduce innovative connection constraints to regulate feature transmission between neurons. Drawing inspiration from this novel concept of neuronal diversity, our model exhibits biological plausibility, facilitating more effective emotion recognition of multisensory information. Experiments on the RAVDESS and eNTERFAVE'05 datasets show that the NDIM achieves the best accuracy of 99.63% and 98.45%, respectively, demonstrating the potential of neuronal-diversity-inspired approaches in advancing multisensory interaction and emotion recognition.

## 1 Introduction

Multisensory emotion recognition is an emerging technique that demonstrates superior performance compared to unisensory recognition due to its ability to mitigate non-robustness observed in unisensory recognition (Sun et al., 2008). Thus this technique has gained significant attention across diverse fields such as human-computer interaction (Abdullah et al., 2021), emotion regulation (Liu et al., 2019), and the diagnosis of emotion-related diseases (Widge et al., 2018). To further enhance emotion recognition performance, multisensory fusion is a crucial approach that effectively exploits complementary information and accentuates the most relevant details (Ranganathan et al., 2016).

With the advancements in deep learning, an increasing number of methods have been employed for fusing multisensory information. Diverse approaches, including Convolutional Neural Network (CNN) based, Recurrent Neural Network (RNN) based, and hybrids with other algorithms, have been employed to accomplish fusion, yielding promising outcomes in multisensory emotion recognition tasks (Zhang et al., 2017; Tzirakis et al., 2017; Tang et al., 2017; Huan et al., 2021). Furthermore, deep generative models, such as variational autoencoders (VAEs) (Zhou et al., 2018; Wang et al., 2022), and generative adversarial networks (GAN) (Luo et al., 2019; Ma et al., 2022), exhibit outstanding performance due to their superior feature representation capabilities (Suzuki & Matsuo, 2022). Nevertheless, the majority of these methods fail to consider the interaction between different senses, a crucial aspect in multisensory fusion and recognition (Mansouri-Benssassi & Ye, 2020b).

In contrast to the aforementioned approaches, large language models like $MulT$ (Tsai et al., 2019), which are based on Transformer architecture, facilitate cross-sensory interaction through diverse attention mechanisms. However, concerns persist regarding their dependence on extensive computation and large datasets, resulting in diminished computational efficiency and elevated training costs (Shin et al., 2022). Given the limitations of prior methods, there is an urgent need for a multisensory recognition model that enables intricate cross-sensory interaction while minimizing computation and data requirements.

The human brain possesses an innate and remarkable ability to perceive and recognize the external environment by efficiently utilizing multisensory information, including vision and hearing (McDonald et al., 2001; Ohshiro et al., 2011). To provide machines with similar advantages, the

*Convergence* and the *Enhancement* model are proposed for effective emotion recognition from multisensory information. However, these models have been shown to be inadequate in achieving cross-sensory interaction. The superior multisensory emotion recognition abilities of the human brain are underpinned by the cross-sensory interaction, as evidenced by cognitive neuroscience research (Alvarado et al., 2007). During this interaction, information from different senses can complement and mutually influence one another, facilitating efficient and comprehensive recognition (Holmes, 2007). Specifically, the multisensory interaction of the human brain is attributed to diverse neurons, including both unisensory and multisensory neurons (Stevenson et al., 2014). These neurons exhibit distinct responses to sensory inputs, with unisensory neurons specifically responding to single sensory information, while multisensory neurons respond to inputs from multiple senses (Stein & Stanford, 2008). The human brain effectively captures cross-sensory interaction due to these neurons' distinct response characteristics, thereby enabling superior multisensory emotion recognition abilities (Laurienti et al., 2005).

Based on the facilitation of diverse neurons, including unisensory and multisensory neurons, in multisensory emotion recognition of the human brain, we propose the Neuronal Diversity Inspired Model(NDIM) for multisensory emotion recognition through Spiking Neural Networks (SNN). Our model incorporates the novel concept of neuronal diversity, making it biologically plausible and enabling more effective multisensory emotion recognition. This research presents several significant contributions, summarized as follows. Firstly, in alignment with the observed neuronal diversity in the human brain, the NDIM incorporates both unisensory and multisensory neurons to effectively model and learn cross-sensory interaction. Secondly, pioneering special connection constraints are designed to regulate feature transmission within the NDIM, reflecting the different response characteristics of diverse neurons. Thirdly, we evaluate the NDIM on the RAVDESS and eNTERFAVE'05 datasets, showing that the NDIM achieves the best accuracy of 99.63% and 98.45%, respectively, consistently outperforming state-of-the-art brain-inspired approaches.

## 2 RELATED WORK

In the context of multisensory emotion recognition, some brain-inspired models have been proposed to effectively recognize the information from multiple senses. The *Convergence* model (Benssassi & Ye, 2023) draws inspiration from the convergence theory, which posits that information from distinct senses converges in higher-order brain regions where it is fused and recognized (Stein & Meredith, 1993). In this model, a convergence layer is built to recognize the concatenated features extracted from the visual and auditory senses, serving as the higher-order region. Before the convergence layer, two separate layers extract sense-specific features. Unfortunately, this model lacks interaction between the individual sensory inputs, leading to suboptimal performance.

The *Enhancement* model (Benssassi & Ye, 2023) is inspired by the enhancement theory that highlights the impact of visual information on auditory cortex activity (Molholm et al., 2002; Jessen & Kotz, 2013). In this model, the auditory feature extraction layer receives inputs not only from the auditory input layer but also from the visual layer. Regrettably, the model solely accounts for unidirectional connections from the visual sense to the auditory sense, thus falling short of achieving comprehensive interaction between the visual and auditory senses.

The *Synch-Graph* model (Mansouri-Benssassi & Ye, 2020a) considers the interaction, incorporating the concept of neural synchrony. Neural synchrony refers to the simultaneous neural oscillations of distinct groups of neurons connected by synapses, and it is considered a facilitator of multisensory interaction (Stein, 2012). In the *Synch-Graph* model, bidirectional connections are established between visual and auditory neurons, and a graph is employed to represent the neural synchrony among these neurons. Graph Convolutional Networks (GCN) are utilized to classify the graphs. However, this model exhibits limited robustness in the presence of noise.

In summary, the first two brain-inspired models have proven inadequate in achieving multisensory interaction. Although the *Synch-Graph* model enables interaction, it comes at the expense of limited robustness. Consequently, there is a pressing need to explore novel brain mechanisms to serve as inspiration for developing a model capable of effectively achieving interaction and emotion recognition.

## 3 NEURONAL DIVERSITY OF THE HUMAN BRAIN

In the human brain, efficient processing and recognition of multisensory emotion rely on diverse neurons that exhibit distinct responses to sensory inputs. This section provides an introduction to these neurons, their response characteristics, and their role in facilitating cross-sensory interaction.

Superior Colliculus (SC) (Cuppini et al., 2011b), the posterior Superior Temporal Sulcus and Gyrus (STS/STG) (Engel et al., 2012; Chabrol et al., 2015), are responsible for multisensory emotion recognition. Within these regions, a variety of neurons, including multisensory neurons and unisensory neurons, are prevalent. Multisensory neurons are defined as neurons that respond to stimuli from more than one sense, while unisensory neurons respond exclusively to a single sense (Fetsch et al., 2013). Unisensory neurons can be further categorized into visual-specific and auditory-specific neurons (Stevenson et al., 2014). Multisensory neurons demonstrate significantly greater responses to multisensory stimuli that share a common source compared to any single sense stimuli (Cuppini et al., 2011a). On the other hand, unisensory neurons exhibit no significant changes in response when presented with multisensory stimuli versus single-sensory stimuli. Due to these response characteristics, unisensory neurons and multisensory neurons exhibit different levels of activation when the human brain receives multisensory input (Stein & Stanford, 2008). Moreover, When a neuron receives and activates in response to information, it selectively transmits this information to target neurons based on its type. As a result, information from different senses can be directionally transmitted between these neurons, influencing and complementing each other, thereby achieving cross-sensory interaction and facilitating multisensory emotion recognition (Allman et al., 2009).

## 4 PROPOSED NDIM MODEL

Building upon the advantages of neuronal diversity in cross-sensory interaction, we present our model in Figure 1. The NDIM consists of three modules: the Unisensory Processing Module, the Neuronal Diversity Module, and the Interaction Module. The Neuronal Diversity Module determines neuron types based on their spiking patterns and devises unique connection constraints to facilitate multisensory interaction and emotion recognition in the Interaction Module.

In the subsequent sections, we present the overarching framework of our model (Section 4.1), followed by the introduction of its key components: the Neuronal Diversity Module (Section 4.2) and the Interaction Module (Section 4.3).

### 4.1 OVERALL ARCHITECTURE

In this model, we initially process the unisensory data in the Unisensory Processing Module to extract semantic features. These extracted features then converge in the Interaction Module for interaction and emotion recognition. To closely emulate the neuronal diversity in the human brain, we design the Neuronal Diversity Module to enable comprehensive interaction through various types of neurons and specific connection constraints.

We investigate two sensory modalities in this study: visual (denoted by the superscript '$(v)$') and auditory (denoted by the superscript '$(a)$'). The input data for each sense is denoted as $\boldsymbol{X}^{(v)} \in \mathbb{R}^{t^{(v)} \times d^{(v)}}$ and $\boldsymbol{X}^{(a)} \in \mathbb{R}^{t^{(a)} \times d^{(a)}}$ for every sample. Here, $t^{(*)}$ and $d^{(*)}$ represent the time dimension and feature dimension, respectively. Initially, the input data undergoes preprocessing and feature extraction, resulting in the primary features of each modality, denoted as $\boldsymbol{F}_p^{(v)}$, and $\boldsymbol{F}_p^{(a)}$. These primary features are then further processed as semantic features, denoted as $\boldsymbol{F}_s^{(v)}$ and $\boldsymbol{F}_s^{(a)}$, through a spiking convolution layer and a pooling layer. Subsequently, the multisensory features $\boldsymbol{F}_s^{(m)}$ are formed by concatenating the semantic features from the visual and auditory senses. The multisensory features are then passed through the Interaction Module for further recognition. The Interaction Module consists of two hidden layers and a readout layer. The hidden layers comprise three types of neurons: unisensory neurons for the visual sense, unisensory neurons for the auditory sense, and multisensory neurons. Specific constraints are designed to govern the connections between different types of neurons, regulating the transmission of features across these layers.

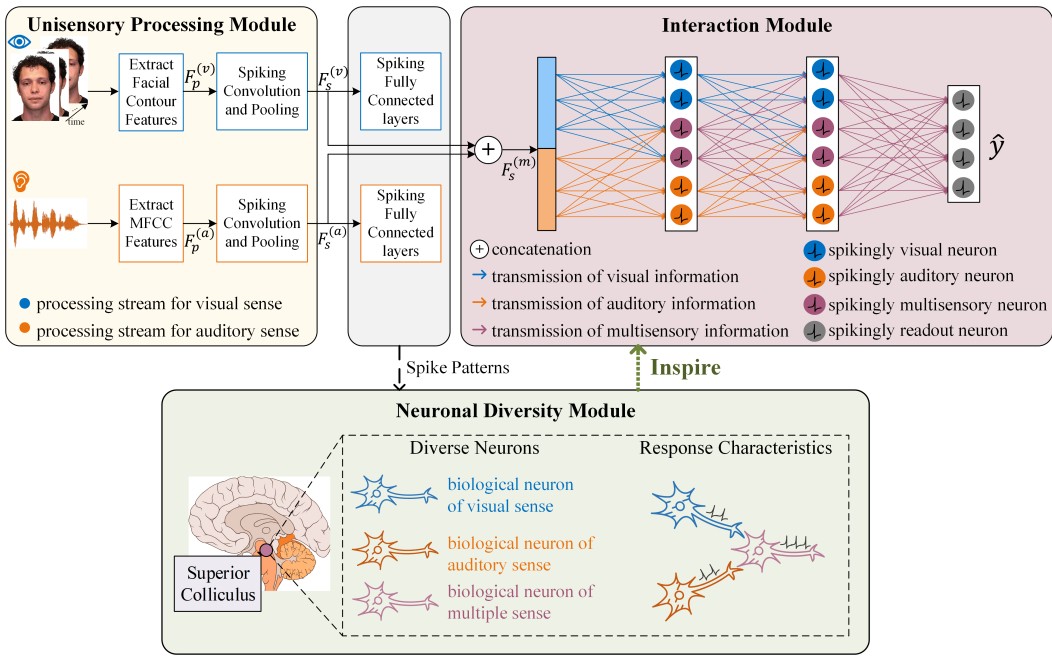

Figure 1: The overall architecture of the proposed NDIM model that is inspired by the neuronal diversity. Three modules are included in the NDIM: the Unisensory Processing Module, the Neuronal Diversity Module, and the Interaction Module. The gray rectangular area accommodates two networks responsible for unisensory emotion recognition. Once trained, the Neuronal Diversity Module receives the spiking patterns of the internal neurons from these networks. The Neuronal Diversity Module determines the neuronal types based on their spiking patterns and designs unique connection constraints to enable the Interaction Module to achieve multisensory interaction and emotion recognition. Notably, each layer of the Interaction Module comprises distinct neuron types. To achieve a unique transmission of sensory information between these neurons, the weights and connection constraints undergo Hadamard product computations.

The neuron model employed in the NDIM architecture is the classical leaky integrated-and-fire (LIF) model. To enhance the learning capacity of the network, neuronal plasticity (Jia et al., 2021) is considered. In this regard, the neuron model incorporates an adaptive firing threshold determined by an ordinary differential equation. For instance, the update of the membrane potential for the $i$-th neuron in the $l$-th hidden layer can be illustrated as follows:

$$C\frac{dV_i(t)}{dt} = g\left(V_i(t) - V_1\right)\left(1 - S_i(t)\right) + \sum_{j=1}^{n^l}\left(W_{i,j}^{(l)}\right)\boldsymbol{F}_s^{(m)} - \gamma a_i(t) \tag{1}$$

$$if\left(V_i(t) = V_{th}\right), \left\{\begin{array}{l} V_i(t) = V_2, \\ S_i(t) = 1, \end{array}\right. \tag{2}$$

$$\frac{da_i(t)}{dt} = (\alpha - 1)a_i(t) + \beta S_i(t) \tag{3}$$

where $C$ is the capacitance parameter, $g$ is the conductance value, $V_i(t)$ is the membrane potential of neuron $i$ at timing $t$, $S_i(t)$ is the firing flag, $V_1$ is the resting potential, $V_2$ is the reset membrane potentials, $V_{th}$ is the firing threshold. $n^l$ is the number of neurons in the $l$-th hidden layer. $W_{i,j}^{(l)}$ is the synaptic weight from the neuron $i$ to the neuron $j$ in the $l$-th layer. The dynamic threshold $a_i(t)$ is accumulated during the period from the resetting to the membrane potential firing, and as the frequency of firing increases, the threshold also increases, and vice versa.

## 4.2 Neuronal Diversity Module

In Section 3, we have explained the significance of different types of neurons and synaptic connections in achieving effective multisensory interaction and emotion recognition. Drawing inspiration from this, our model incorporates diverse neurons and special weight constraints to facilitate comprehensive interaction among multisensory information. To extract this information, we propose the Neuronal Diversity Module, as illustrated in Figure 2. This subsection details the process of determining diverse neurons and establishing two connection constraints based on them. These constraints aim to facilitate multisensory interaction by enforcing specific weight configurations.

To identify diverse neurons and establish connection constraints, additional unisensory emotion recognition networks ($Re^{(v)}$, $Re^{(a)}$) are designed, as shown in the gray rectangular area of Figure 1. These networks consist of two fully connected layers and one readout layer, serving to classify the unisensory features $\boldsymbol{F}_p^{(v)}$, and $\boldsymbol{F}_p^{(a)}$. The ensemble comprising the unisensory processing and recognition network for each single sense can be trained separately. Subsequently, the spiking patterns of neurons from the trained recognition networks $Re^{(v)}$ and $Re^{(a)}$ are recorded and utilized by the Neuronal Diversity Module to identify the diverse neurons and establish connection constraints.

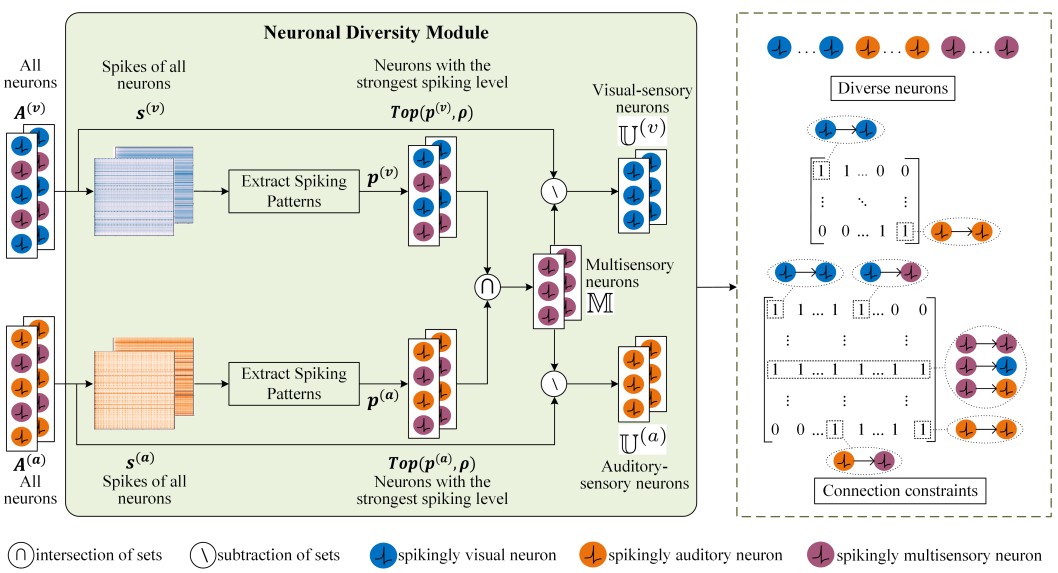

Figure 2: The detailed diagram of the Neuronal Diversity Module. $\mathbb{M}$ represents the multisensory neurons and $\mathbb{U}$ represents the unisensory neurons.

### 4.2.1 Determining Diverse Neurons

To elaborate on our ideas of these modules, we explain how the NDIM correlates with multisensory emotion recognition mechanisms in the human brain. Unisensory information is processed through feature extraction and recognition processes that result in a final classification. This process corresponds to the unisensory information recognition circuit in the human brain, with the visual and auditory processing streams corresponding to the ventral visual and auditory pathways, respectively. The architectures of $Re^{(v)}, Re^{(a)}$, and the Interaction Module are identical since they mimic higher-order regions. However, the difference lies in the populations of neurons they mimic. Unisensory neurons for vision and multisensory neurons respond to visual stimuli, which corresponds to the neurons that $Re^{(v)}$ mimics, while $Re^{(a)}$ involves unisensory neurons for audio and multisensory neurons. Therefore, different activation patterns of neurons can differentiate between unisensory and multisensory neurons in higher-order regions.

The Neuronal Diversity Module identifies different types of neurons in a layer-by-layer fashion. In particular, the spiking patterns of neurons in the first layer of $Re^{(v)}, Re^{(a)}$ are used to identify neurons in the first layer of the Neuronal Diversity Module, and this process is repeated for each

subsequent layer. To represent spike patterns, $\boldsymbol{S}_i^{l,(v)}, \boldsymbol{S}_i^{l,(a)} \in \mathbb{R}^{n^l \times t'}$ denote the spikes of all neurons in the $l$-th layer, for each sample of visual and auditory senses. Spike patterns, denoted as $\boldsymbol{P}^{l,(v)}$ and $\boldsymbol{P}^{l,(a)}$, are obtained by averaging spikes on a timing scale and samples, which are formulated as:

$$\boldsymbol{P}^{l,(v)} = \frac{1}{n^l} \sum_{i=1}^{n^l} Mean_t \left( \boldsymbol{S}_i^{l,(v)} \right), \boldsymbol{P}^{l,(a)} = \frac{1}{n^l} \sum_{i=1}^{n^l} Mean_t \left( \boldsymbol{S}_i^{l,(a)} \right) \tag{4}$$

where $Mean_t(\cdot)$ represents the average value on timing scale. Then, the multisensory neurons are determined by,

$$\mathbb{M}^l = Top \left( \boldsymbol{P}^{l,(v)}, \rho \right) \cap Top \left( \boldsymbol{P}^{l,(a)}, \rho \right) \tag{5}$$

where $Top(\cdot, \rho)$ is used to identify neurons with the highest spiking level based on their spike patterns. Here, $\rho$ is a hyperparameter ranging from 0 to 1, indicating the percentage of neurons with the top $\rho$ highest spikes that are considered to have the strongest spiking level. Multisensory neurons in the $l$-th layer, denoted as $\mathbb{M}^l$, are found by intersecting the visual neurons with the strongest spiking level and the neurons associated with auditory modality. Unisensory neurons, on the other hand, are obtained by taking the difference between all neurons and the multisensory neurons in the $l$-th layer, which is identified by,

$$\mathbb{U}^{l,(v)} = \mathbb{A}^{l,(v)} \backslash \mathbb{M}^l, \mathbb{U}^{l,(a)} = \mathbb{A}^{l,(a)} \backslash \mathbb{M}^l \tag{6}$$

where the $\mathbb{U}^{l,(v)}$ and $\mathbb{U}^{l,(a)}$ represent unisensory neurons in the $l$-th layer, $\mathbb{A}^{l,(v)}$ and $\mathbb{A}^{l,(a)}$ represent all neurons in the $l$-th layer.

### 4.2.2 ESTABLISHING WEIGHT CONSTRAINTS

Inspired by the response characteristics of different neurons, we establish connection constraints, denoted as matrices $\boldsymbol{C}^l \in \mathbb{R}^{n'^{l-1} \times n'^l}$, where $n'^l$ represents the number of neurons in the $l$-th layer of the Neuronal Diversity Module. These matrices facilitate the interaction between neurons of different senses. They are filled with 1 or 0, representing connected and disconnected weights, respectively. The function $D(i,j)$ determines the connection constraint from the $i$-th neuron in the $l-1$th layer (source neuron) to the $j$-th neuron in the $l-1$th layer (target neuron).

The function value of $D(i,j)$ is determined by the following rules. Firstly, if the source neuron is unisensory and the target neuron is either unisensory of the same modality or multisensory, the function value is set to 1. Secondly, if the source neuron is multisensory and the target neuron is unisensory, the function value is set to 1. Thirdly, In all other cases, the function value is set to 0. These rules define the masks for both layers. The weight constraints, along with the diverse neurons, are then projected to the Interaction Module for multisensory emotion recognition.

### 4.3 INTERACTION MODULE

The Interaction Module aims to enhance multisensory emotion recognition by leveraging diverse neurons and connection constraints introduced earlier. This subsection provides a detailed explanation of how the module achieves interaction and emotion recognition based on these diverse neurons and weight constraints.

Diverse neurons and weight constraints are determined and established to guide the Interaction Module for further recognition. The diverse neurons facilitate the identification of multisensory and unisensory neurons within the module. Meanwhile, the weight constraints restrict the connections between neurons through the Hadamard product with $\boldsymbol{C}^l$. This approach aligns with the response characteristics of neuronal diversity, where multisensory neurons in higher-order regions respond to stimuli from multiple senses, while unisensory neurons only respond to stimuli from the same sense.

Through these neurons and constraints, interactions between different senses are achieved. Specifically, the auditory features $\boldsymbol{F}_s^{(a)}$ are connected to the multisensory neurons of the first layer $\mathbb{M}^1$,

which, connect to the visual neurons in the second layer $\mathbb{U}^{2,(v)}$. This configuration enables a unidirectional projection from the auditory sense to the visual sense, as well as vice versa. Consequently, the interaction between the visual and auditory senses is achieved.

## 5 EXPERIMENTS

### 5.1 EXPERIMENTS SETUP

#### 5.1.1 DATASETS.

Two datasets were used to evaluate our model. The first dataset is the Ryerson Audio-Visual Database of Emotional Speech and Song (RAVDESS) (Martin et al., 2006). It includes recordings from 24 participants, with a balanced gender distribution. The participants read a sentence in eight different emotional states: neutral, calm, happy, sad, angry, fearful, disgust, and surprised. For this study, we focused solely on the speech and video modalities within this dataset. The second dataset, eNTERFACE'05 (Livingstone & Russo, 2018), consists of recordings from 42 participants. The participants comprise 81% male and 19% female. The audio recordings have a sampling rate of 48000Hz in 16-bit format, while the videos have a frame rate of 25 frames per second. Each participant expressed six different emotions: anger, disgust, fear, happiness, sadness, and surprise.

#### 5.1.2 FEATURE EXTRACTION.

The primary features of each sense, denoted as $\boldsymbol{F}_p^{(v)}$, and $\boldsymbol{F}_p^{(a)}$, are obtained by feature extraction. For the visual modality, 15 frames are extracted at equal intervals from each video. Facial contours are then extracted from these frames and downscaled to a size of 28x28 pixels, serving as visual features for each frame. Consequently, the final dimension of the visual feature $\boldsymbol{F}_p^{(v)}$ for each video is $\boldsymbol{R}^{15*784}$. Regarding the auditory modality, we extract Mel-scale Frequency Cepstral Coefficients (MFCC) as auditory features $\boldsymbol{F}_p^{(a)}$ for each speech sample. MFCC is a widely used feature in speech recognition. The average number of frames across all speech samples is 280, with a feature dimension of 12 per frame. Therefore, the final dimension of the auditory feature is $\boldsymbol{R}^{12*280}$.

### 5.2 OVERALL PERFORMANCE

We focus on evaluating the performance of the NDIM modal in multisensory emotion recognition by conducting a series of experiments on two datasets. We compare our proposed model with state-of-the-art techniques and analyze its interpretability. Tables 1 and 2 show the performance comparison between the NDIM modal and the state-of-the-art baselines on the two datasets.

Table 1: Comparison of accuracy for multisensory emotion recognition on RAVDESS dataset. $^*$ represents that the performance of this model is obtained from the relevant paper.

| Model | Neutral | Clam | Happy | Sad | Angry | Fearful | Disgust | Surprised | Acc |
|---|---|---|---|---|---|---|---|---|---|
| *Convergence** | - | - | - | - | - | - | - | - | 0.8130 |
| *Enhancement** | - | - | - | - | - | - | - | - | 0.7330 |
| *Synch-Graph** | | | 1.0000 | 1.0000 | 1.0000 | 0.9550 | 0.9310 | 0.9290 | 0.9830 |
| *MR-SNN* | 0.9655 | 0.9825 | 1.0000 | 0.9355 | 0.9310 | 0.9091 | 0.9474 | 0.9655 | 0.9537 |
| *MulT** | - | - | - | - | - | - | - | - | 0.7416 |
| **NDIM** | **1.0000** | **1.0000** | **1.0000** | **0.9933** | **1.0000** | **0.9871** | **1.0000** | **0.9931** | **0.9963** |

Our model achieves the best performance on both datasets, with an accuracy of 99.63% on the RAVDESS dataset and 98.45% on the eNTERFACE'05 dataset. Notably, these outcomes outperform the state-of-the-art multisensory emotion recognition method employed on the same datasets.

Among the baselines, the *Synch-Graph* model performs best on the two datasets, achieving accuracies of 98.30% and 96.82% respectively. This model learns synchrony patterns between audio and visual neuron groups. In comparison, our model recognizes concatenated features, drawing inspiration from neuronal diversity. Interaction is achieved through special connections between

Table 2: Comparison of accuracy for multisensory emotion recognition on eNTERFACE'05 dataset. * represents that the performance of this model is obtained from the relevant paper.

| Model | Angry | Disgust | Fear | Happy | Sad | Surprised | Acc |
|---|---|---|---|---|---|---|---|
| *Convergence** | - | - | - | - | - | - | 0.8330 |
| *Enhancement** | - | - | - | - | - | - | 0.8330 |
| *Synch-Graph** | 0.9470 | 0.9550 | 1.0000 | 1.0000 | 0.9200 | 1.0000 | 0.9682 |
| *MR-SNN* | 0.9231 | 0.9180 | 0.9032 | 0.9118 | 0.9231 | 0.8955 | 0.9124 |
| **NDIM** | **0.9524** | **1.0000** | **1.0000** | **1.0000** | **0.9524** | **1.0000** | **0.9845** |

unisensory neurons and multisensory neurons. The Neuronal Diversity Module determines neuron diversity and establishes connection constraints, enabling the Interaction Module to interact and recognize multisensory information. Owing to these advantages, our model achieves an accuracy of 99.63% with eight classes on the RAVDESS dataset, while the $Synch\text{-}Graph$ achieves 98.30% accuracy with six classes. The two classes not considered by the $Synch\text{-}Graph$ model are *neutral* and *calm*. In these two classes, the ND-MRM model achieves 100% accuracy. Furthermore, our model outperforms the $Synch\text{-}Graph$ by 1.63% on the eNTERFACE'05 dataset. Compared with other brain-inspired methods such as the $Convergence$, $Enhancement$, and $M\text{-}SNN$, our model outperforms them by 18.33%, 26.33%, 4.26% on the RAVDESS dataset and 15.15%, 15.15%, 7.21% on the eNTERFACE'05 dataset respectively.

In addition to these brain-inspired methods, we also compare the performance of the $MulT$ model with our model on the RAVDESS dataset. The accuracy of the $MulT$ model on seven classes is 74.16%, while our model outperforms $MulT$ by 25.47% (Chumachenko et al., 2022).

In summary, our model demonstrates superior performance on both datasets for the task of multisensory emotion recognition. The NDIM model captures the interaction between different senses using diverse neurons and special connection constraints, outperforming the other brain-inspired methods and the $MulT$ model's pairwise cross-modal attention approach.

## 5.3 Ablation Study

In this subsection, we conduct two ablation experiments to further investigate the impact of neuronal diversity on multisensory emotion recognition. Firstly, we examine the influence of neuronal type on emotion recognition in the hidden layers of the Interaction Module. We compare models with either all unisensory neurons or all multisensory neurons to understand whether a model with multiple types of neurons working together outperforms a model with a single type of neurons when the total number of neurons remains constant. Secondly, we study the effect of varying the hyperparameter $\rho$, which determines the number of neurons, on multisensory emotion recognition.

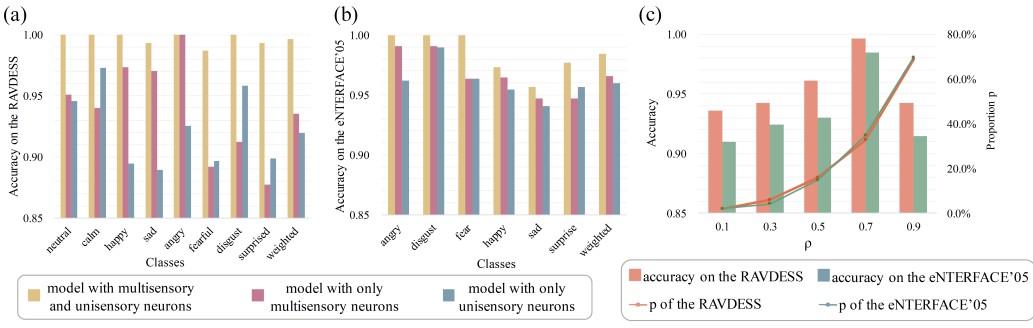

Figure 3: Accuracy comparison of the ablation experiments on neuronal diversity. From left to right: (a) Accuracy comparison of neuronal types on the RAVDESS dataset. (b) Accuracy comparison about neuronal types on the eNTERFACE'05 dataset. (c) Accuracy comparison of neuronal numbers on the two datasets.

### 5.3.1 ABLATION STUDY ON NEURONAL TYPES

Both multisensory neurons and unisensory neurons are components of the NDIM model, which aims to achieve cross-sensory interaction. Therefore, this ablation experiment investigates the performance of models that contain either only multisensory neurons or only unisensory neurons in the two hidden layers. The performance comparison is shown in Figure 3(a)(b).

Compared with the other two types of models, the model containing only unisensory neurons exhibits the worst performance, achieving 91.94% on the RAVDESS dataset and 95.97% on the eNTERFACE'05 datasets. This is evident because unisensory neurons solely respond to a single sensory stimulus, preventing the model from achieving cross-modal interaction and resulting in its inferior performance. Additionally, the model containing only multisensory neurons surpasses those with unisensory neurons but is still outperformed by the model with multiple types of neurons. Unlike the NDIM, which achieves regular interaction of multisensory information through neuron diversity and connection constraints, the hidden layers in the model that contain only multisensory neurons are fully connected. This distinction accounts for our model's superiority.

### 5.3.2 ABLATION STUDY ON THE NUMBER OF NEURONS

As mentioned in Section 4, the interaction between different senses is facilitated by distinct types of neurons and their synaptic connections, which are determined by the hyperparameter $\rho$. Figure 3(c) presents the values of three parameters: $\rho$, the proportion ($p$) of multisensory neurons among all neurons, and the weighted accuracy on the RAVDESS and eNTERFACE'05 datasets.

As shown in the figure, as the hyperparameter $\rho$ approaches 0, the number of multisensory neurons decreases. When $\rho$ is 0.1, the number of multisensory neurons becomes almost zero, leading to a near disappearance of the interaction between multiple senses. In this condition, the model achieves a weighted accuracy of 93.61% and 91.25% on the two datasets, respectively, which is significantly lower than the accuracy achieved in other conditions. Moreover, when $\rho$ is set to 0.9, the number of multisensory neurons accounts for over 67% of the total number of neurons. The performance in this condition is lower than that in conditions where the number of multisensory neurons accounts for approximately 33%. We attribute this to an imbalance between the number of unisensory neurons for auditory and visual senses and the number of multisensory neurons. When $\rho$ is 0.9, the model overly emphasizes the interaction between different senses while neglecting the extraction of higher-level features from the individual senses. Therefore, when $\rho$ is 0.7, the number of multisensory neurons accounts for approximately 33%, and the number of different types of neurons reaches a relative equilibrium, resulting in the best performance of our model.

## 6 CONCLUSION

This study aims to develop a novel multisensory emotion recognition model called NDIM, drawing inspiration from the neuronal diversity observed in the human brain. By incorporating both unisensory and multisensory neurons, our model effectively captures cross-sensory interactions, facilitating multisensory emotion recognition. Additionally, we implement special connection constraints to regulate feature transmission, which aligns with the distinctive response characteristics exhibited by diverse neurons. The performance of the NDIM on the RAVDESS and eNTERFAVE'05 datasets is 99.63% and 98.45%, respectively, revealing its superiority over alternative brain-inspired approaches.

For future research, we plan to extend the application of our method to multiple other sensory modalities, exploring its effectiveness and adaptability in various tasks. Additionally, we aim to explore the development of an end-to-end model that integrates all stages of information processing within a unified framework.

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

## A  BASELINES.

We select a set of brain-inspired methods as our baselines, and they are known for their superior performance. (1) The $Convergence$ (Benssassi & Ye, 2023) model is inspired by the convergence theory with a convergence layer to integrate concatenated features of different senses. In this model, there is no interaction between the visual and auditory senses. (2) The $Enhancement$ (Benssassi & Ye, 2023) model is inspired by the enhancement theory with unidirectional connection from visual neurons to auditory neurons. In this model, the interaction between the senses has not been achieved. (3) The $Synch\text{-}Graph$ (Mansouri-Benssassi & Ye, 2020a) model realizes the interaction, inspired by neural synchrony. The synchrony is captured by a graph network and then recognized by GCN. (4) The $MR\text{-}SNN$ (Jia et al., 2022) model is proposed to recognize digits in MNIST and TIDigits datasets. It's based on the motifs which are weight connections between neurons and are extracted from pre-trained networks used in unisensory tasks. (5) The $MulT$ (Tsai et al., 2019) adopts directional pairwise cross-modal attention, which attends to the interaction between multimodal sequences, to recognize information in an end-to-end manner.

## B  CONFIGURATIONS

In the spiking convolution layer, the number of channels is 4 and the kernel size is $5 * 5$. Followed by an average pooling layer. The $Re^{(v)}$ and $Re^{(v)}$ are composed of two fully connected layers ($n^l$ is 200), and an output layer with the same number of labels of each dataset. After the determination of neuronal types, the number of the $l$-th hidden layer in the Interaction Module is the sum of three components: multisensory neurons in the $l$-th layer $\mathbb{M}^l$, unisensory neurons $\mathbb{U}^{l,(v)}$ and $\mathbb{U}^{l,(a)}$.

The hyperparameter $\rho$ is initially set to 0.7. The capacitance $C$ is $1\mu F/cm^2$, conductivity $g$ is $0.2nS$, time constant is $1ms$, resting potential $V_1$ is equal to reset potential $V_2$ with $0mV$. The firing threshold is $0.5mV$ in the beginning. For the adaptive threshold, we set $\alpha$=0.9, $\beta$=0.1, and $\gamma$=1.

