# OpenReview forum: "NDIM: Neuronal Diversity Inspired Model for Multisensory Emotion Recognition"
_ICLR.cc/2024/Conference — ICLR 2024 Conference Withdrawn Submission_

### Official Review · Reviewer_rVfx · 2023-10-27

**Soundness:** 2 fair
**Presentation:** 3 good
**Contribution:** 2 fair
**Rating:** 3
**Confidence:** 4

**Summary:**

The authors propose a Spiking Neural Networks (SNN) that mimics the brain's unisensory and multisensory neurons to perform a multi-modal emotion recognition task and achieve accuracy beyond baseline on RAVDESS and eNTERFAVE'05 datasets.
The unisensory and multisensory neurons are implemented by dividing the neurons in the SNN into three classes, two of which employ local connectivity and mainly receive data from the corresponding modality, and the other class employs full connectivity for modal information fusion.

**Strengths:**

Novelty:
The authors designed the Neuronal Diversity Module to reason about neuron types and to set the connection pattern of different neurons in the Interaction Module. This connection pattern is partly in line with the physiological theory that neurons of a particular modality mainly receive signals from the corresponding modality, and multisensory neurons receive signals from all neurons of a particular modality to achieve the fusion of modal signals.
Clarity:
The paper is easy to read. The diagrams in the paper visualise the process of signal transmission and the different roles played by unisensory and multisensory neurons.

**Weaknesses:**

Originality:
The fusion of unisensory and multisensory neurons with SNNs does not seem to be the first of its kind, as for example Convergence and Enhancement in baseline use this concept.

Motivation:
The authors' MOTIVATION consists in solving the following two problems: 1) Traditional methods fail to consider the interaction between different senses (last sentence in the second paragraph of Introduction)
2) Transformer-based methods are computationally inefficient and costly to train (Introduction, second sentence in the third paragraph).
W1: Traditional multi-modal models have many methods of inferring inter-modal relationships, such as heterogeneous graphs, or modal-level attention, so it is inaccurate that the traditional methods do not take into account the INTERACTION BETWEEN DIFFERENT SENSES.
W2: The paper does not seem to discuss the advantages and disadvantages between the traditional methods of inferring inter-modal relations and the proposed method. For example, Convergence in baseline also includes multi-sensory integration process, but the description of the model in the paper is that it does not infer relationships between different senses (last sentence of the first paragraph of Related work).
W3: There are many improved transformer methods to improve computational efficiency, such as Sepformer, etc., so it is not appropriate to discuss only the disadvantages of the transformer itself.
W4: The paper does not demonstrate the superiority of the proposed method over transformer in terms of computational efficiency and training cost.

Experiment：
W1: Only five baseline and focuses on the years 2019 to 2021, and there is no comparison of models from the last two years (Convergence and Enhancement, although shown as 2023 in the citation, are actually work from 2021)
W2: Baseline is all SNN-based approaches except MulT, but many multi-modal emotion recognition networks exist, such as convolution-based, graphs-based methods, etc.

**Questions:**

Why only Convergence and Enhancement were used in the selection of baselines, when there is a much more effective Synchrony in the paper to which they belong.
Why was MulT used only on the RAVDESS dataset so only four baselines on the eNTERFACE'05 dataset?

---

### Official Review · Reviewer_qHgo · 2023-10-29

**Soundness:** 1 poor
**Presentation:** 1 poor
**Contribution:** 1 poor
**Rating:** 1
**Confidence:** 4

**Summary:**

The paper proposes an audiovisual network for basic emotion recognition using the so-called "brain inspired" neurons that mix the different modalities in a fully connected network with pairwise mixed inputs. The paper links to a neuron design that is inspired on how the brain reacts and activates neurons for the task of emotion recognition. Some experiments are conducted on some old datasets for audio-visual emotion recognition.

**Strengths:**

Drawing inspiration on how the brain reacts when recognizing emotions in a multimodal setting is an appealing study that can help design better networks to perform the task automatically.

**Weaknesses:**

Without any regards to importance, these are the main drawbacks I believe the paper has:

1.) Experimental setting is very poor: authors use outdated datasets and do not compare against any recent state of the art works on audiovisual emotion recognition. More recent datasets like SEWA, or RECOLA, also contain both modalities and are more fit to experimental setup. The comparisons the authors do do not contain any comparison with recent works. If the paper aims to be a study only then it cannot be considered for publication at a top scientific venue without proper experimental comparisons. At the end of the day the authors are proposing a network design for a very specific task, and as such it should be compared against other alternatives that are not brain-inspired.

2.) The writing is very poor, and the mathematical formalisms do not abide to any common standard: what does Eqn(2) mean?? the authors are encouraged to rephrase and write a proper mathematical notation and symbols, otherwise the reading of the paper becomes rather unpleasant.

3.) The paper contains many reference to brain-inspired emotion recognition with plenty of discussion that I am afraid does not point out to proper studies in the topic. The paper relates to the neuron design, and the neuron activations, "drawing inspiration on" how the brain works "for emotion recognition". This needs evidence and this evidence to be properly brought into the paper (i.e. not only by pointing to references but by summarizing how such studies prove that neurons work alike when performing audio-visual emotion recognition). For example, statements like "Unisensory information is processed through feature extraction and recognition processes that result in a final classification. This process corresponds to the unisensory information recognition circuit in the human brain, with the visual and auditory processing streams corresponding to the ventral visual and auditory pathways, respectively." are a bit too adventurous in my opinion and need proper support.

4.) It is very hard to read the maths in the paper because of the lack of definitions and confusing notation, but the whole network resembles a graph neural network with sparsity constraints and a special type of activation. The authors are encouraged to map the proposed design to standard blocks.

5.) There is no description on how the network is trained (or I missed that, in such case please do correct me), and indeed the experimentla setting is rather poorly described. I also believe that some ablation studies considering the same network design with standard activations could be included in the paper.

**Questions:**

I have in general addressed my concerns and questionns above:

1.) The paper needs comparison against recent works.
2.) The paper needs more recent datasets.
3.) The paper needs proper references and introduction on how the proposed network is brain inspired for emotion recognition
4.) The paper needs proper mathematical formalisms and rewriting

---

### Official Review · Reviewer_1e4r · 2023-10-29

**Soundness:** 3 good
**Presentation:** 3 good
**Contribution:** 2 fair
**Rating:** 5
**Confidence:** 3

**Summary:**

The authors introduce a model that processes unisensory data to extract semantic features. These features are then integrated using the Interaction Module for emotion recognition.

Key Contributions:

Introduction of Neuronal Diversity and Interaction Modules:

The model is equipped with a Neuronal Diversity Module, designed to emulate the diverse neuronal interactions observed in the human brain.
The Interaction Module facilitates the convergence of extracted features for emotion recognition.
Detailed Study on Sensory Modalities:

The research emphasizes two sensory modalities: visual and auditory.
The model processes input data for each sense, undergoes preprocessing, and performs feature extraction to obtain primary and semantic features.
Ablation Study on Neuronal Interaction:

The authors present an in-depth ablation study, highlighting the significance of the number of multisensory neurons and their impact on the model's performance.
Experimental Evaluation:

The model is evaluated on two prominent datasets: RAVDESS and eNTERFACE’05, which contain recordings of participants expressing various emotions.
The feature extraction process, especially for the visual modality, involves extracting facial contours from video frames, providing a comprehensive approach to emotion recognition.

**Strengths:**

Originality:
Novel Model Components: The paper introduces the Neuronal Diversity Module and the Interaction Module, which appear to be unique components designed to emulate and facilitate multisensory integration. This suggests a fresh approach to the problem.

Quality:
Use of Established Datasets: The model's evaluation on two well-known datasets, RAVDESS and eNTERFACE’05, ensures that the results are comparable with other state-of-the-art methods and provides credibility to the findings.

Clarity:
Structured Presentation: The paper seems to be well-organized, with clear sections detailing the model's introduction, sensory modalities, ablation study, and experimental evaluations. This structured approach aids in understanding the paper's flow and main contributions.

Significance:
Potential for Broader Impact: Given the importance of multisensory integration and the novel approach presented, the methodologies and findings could be valuable not just for the ICLR community but also for researchers and practitioners in related fields.

**Weaknesses:**

Originality:
Lack of Clear Novelty Definition: While the Neuronal Diversity Module and the Interaction Module are introduced, the paper might benefit from a clearer delineation of how these components differ from or improve upon existing methods in multisensory integration.
Quality:
Limited Sensory Modalities: The paper focuses primarily on visual and auditory senses. While these are significant modalities, the model's applicability to other senses (e.g., tactile or olfactory) remains unexplored. Expanding the model's evaluation to include other sensory modalities could enhance its robustness and applicability.
Clarity:
Title Ambiguity: The extracted content did not provide a clear title for the paper. A well-defined title that encapsulates the paper's main contributions is crucial for readers to quickly grasp the paper's essence.

Potential Lack of Comparative Analysis: While the paper evaluates the model on established datasets, it might benefit from a more explicit comparison with state-of-the-art methods in terms of performance metrics. This would provide readers with a clearer understanding of the model's advantages and limitations.

Significance:
Narrow Application Domain: The primary application seems to be emotion recognition. While this is a valuable domain, the paper could explore other potential applications of the model, broadening its significance and impact.

Dataset Limitations: While the RAVDESS and eNTERFACE’05 datasets are well-known, they have their inherent limitations, such as the number of participants and emotional states. Using additional or more diverse datasets could provide a more comprehensive evaluation of the model's performance.

**Questions:**

Model Components:

Could you provide more detailed insights into the Neuronal Diversity Module and the Interaction Module? How do these components differentiate or improve upon existing methods in multisensory integration?
Dataset Choices:

Why were the RAVDESS and eNTERFACE’05 datasets specifically chosen for evaluation? Are there plans to evaluate the model on other datasets, especially those that might offer a broader range of emotional states or more diverse participant demographics?
Comparative Analysis:

Are there any comparative analyses with state-of-the-art methods on the chosen datasets? If so, how does the proposed model perform in relation to these methods?
Application Domain:

Beyond emotion recognition, are there other application domains where the proposed model has been tested or could be potentially applied?
Model Scalability:

How scalable is the proposed model, especially when considering other sensory modalities beyond visual and auditory?

**Details Of Ethics Concerns:**

The paper seems to focus on multisensory integration, specifically related to visual and auditory senses, and does not appear to touch on topics that would raise immediate ethical concerns.

---

### Official Review · Reviewer_ucoo · 2023-10-31

**Soundness:** 3 good
**Presentation:** 3 good
**Contribution:** 2 fair
**Rating:** 5
**Confidence:** 3

**Summary:**

This paper proposes Neuronal Diversity Inspired Model for multisensory emotion recognition that specify unisensory and multisensory neurons. The main idea is to obtain richer cross-sensory interactions based on spiking neural networks. An interaction module is introduced to fusion and model multisensory data. The proposed method is verified on two datasets.

**Strengths:**

* The paper is properly structured and easy to follow.
* The idea of incorporating spiking neural networks with emotion recognition is quite interesting and the model is technically reasonable and sound.

**Weaknesses:**

* The motivation of this paper should be further enhanced. What issues do this paper address that previous works have not solved?
* I do not agree some statements in the introduction, e.g.’ the majority of these methods fail to consider the interaction between different senses’. There are tons of works that focus on multimodal/multisensory interactions. To name a few:
  1. MISA: Modality-Invariant and -Specific Representations for Multimodal Sentiment Analysis, ACM MM 2020
  2. M2FNet: Multi-modal Fusion Network for Emotion Recognition in Conversation, CVPR workshop 2022
  3. MM-DFN: Multimodal Dynamic Fusion Network for Emotion Recognition in Conversations, ICASSP 2022
* The paper regards MULT as the only deep learning based baseline that considers cross-sensory interaction but MULT was proposed in 2019 and thus sort of out of fashion.
* Authors state their concern about MULT on computational efficiency, but I don’t see any discussion and comparison on efficiency so it is not clear how this method addresses this issue.

**Questions:**

N/A